# The burden of stillbirths in low resource settings in Latin America: Evidence from a network using an electronic surveillance system

**Bremen de Mucio**[1], **Claudio Sosa**[1], **Mercedes Colomar**[1], **Luis Mainero**[1], **Carmen M. Cruz**[2], **Luz M. Chévez**[2], **Rita Lopez**[2], **Gema Carrillo**[3], **Ulises Rizo**[3], **Erika E. Saint Hillaire**[4], **William E. Arriaga**[5], **Rosa M. Guadalupe Flores**[5], **Carlos Ochoa**[6], **Freddy Gonzalez**[7], **Rigoberto Castro**[7], **Allan Stefan**[8], **Amanda Moreno**[9], **Sherly Metelus**[10], **Renato T. Souza**[10], **Maria L. Costa**[10], **Adriana G. Luz**[10], **Maria H. Sousa**[11], **José G. Cecatti**[10]*, **Suzanne J. Serruya**[1]

1 Latin American Center of Perinatology (CLAP-PAHO), Montevideo, Uruguay, 2 Hospital Berta Calderon Roque, Managua, Nicaragua, 3 Hospital España, Chinandega, Nicaragua, 4 Hospital San Lorenzo de Los Mina, Santo Domingo Este, Dominican Republic, 5 Hospital Regional de Ocidente, Quetzaltenango, Guatemala, 6 Hospital San Felipe, Tegucigalpa, Honduras, 7 Hospital Roberto Suazo Cordova, La Paz, Honduras, 8 Hospital Leonardo Martinez Valenzuela, San Pedro Sula, Honduras, 9 Hospital Boliviano Japones, El Alto, Bolivia, 10 Department of Obstetrics and Gynecology, University of Campinas, Campinas, Brazil, 11 Jundiaí School of Medicine—HU/FMJ, Jundiaí, SP, Brazil

* cecatti@unicamp.br

## Abstract

### Objective

To determine stillbirth ratio and its association with maternal, perinatal, and delivery characteristics, as well as geographic differences in Latin American countries (LAC).

### Methods

We analysed data from the Perinatal Information System of the Latin American Center for Perinatology and Human Development (CLAP) between January 2018 and June 2021 in 8 health facilities from five LAC countries (Bolivia, Guatemala, Honduras, Nicaragua, and the Dominican Republic). Maternal, pregnancy, and delivery characteristics, in addition to pregnancy outcomes were reported. Estimates of association were tested using chi-square tests, and $P < 0.05$ was regarded as significant. Bivariate analysis was conducted to estimate stillbirth risk. Prevalence ratios (PR) with their 95% confidence intervals (CI) for each predictor were reported.

### Results

In total, 101,852 childbirths comprised the SIP database. For this analysis, we included 99,712 childbirths. There were 762 stillbirths during the study period; the Stillbirth ratio of 7.7/1,000 live births (ranged from 3.8 to 18.2/1,000 live births across the different maternities); 586 (76.9%) were antepartum stillbirths, 150 (19.7%) were intrapartum stillbirths and

**Data Availability Statement:** The property of data used in this manuscript is of each participating country, coordinated by the PAHO-CLAP in Montevideo, Uruguay. The data can be available

from there upon a reasonable request. Data access requests can be sent to the following email: pahoerc@paho.org. The data from the Hospital SIP´s database (third part property) is used responsibly and in accordance with the ethical guidelines and regulations governing; restricted sharing may be allowed under formal request outlining the intended use.

**Funding:** The author(s) received no specific funding for this work.

**Competing interests:** The authors have declared that no competing interests exist.

26 (3.4%) with an ignored time of death. Stillbirth was significantly associated with women with diabetes (PRadj 2.36; 95%CI [1.25–4.46]), preeclampsia (PRadj 2.01; 95%CI [1.26–3.19]), maternal age (PRadj 1.04; 95%CI [1.02–1.05]), any medical condition (PRadj 1.48; 95%CI [1.24–1.76, and severe maternal outcome (PRadj 3.27; 95%CI [3.27–11.66]).

## Conclusions

Pregnancy complications and maternal morbidity were significantly associated with stillbirths. The stillbirth ratios varied across the maternity hospitals, which highlights the importance for individual surveillance. Specialized antenatal and intrapartum care remains a priority, particularly for women who are at a higher risk of stillbirth.

## Introduction

According to WHO, approximately 2.6 million stillbirths at 28 weeks of gestation or later occurred worldwide in 2015 [1]. However, depending on income status, glaring discrepancies exist in ratios and causes of stillbirths between countries and regions. Fetal death ratios (FDR) in sub-Saharan Africa are around 10-fold higher than FDR in high-income countries (29 vs 3 per 1,000 births, respectively). From 2000 to 2015, the annual stillbirth rate decreased by 2% worldwide. Nevertheless, this decrease was still lower than maternal death ratios (3%) and lower than death rates of children under 5 years of age (3.9%). Most of these deaths are preventable. Improvements in periconceptional and pregnancy healthcare can potentially reduce the majority of stillbirths [2,3]; already existing cost-effective interventions can save the lives of more than one million babies annually worldwide. [2–4].

To achieve the Sustainable Development Goals, the UN added new goals to the Global Health Strategies for Women, Children and Adolescents [5]. These goals focused on neonatal and stillbirths, family planning and adolescent health. The action plan named Every Newborn emerged, aimed at ending preventable stillbirths by 2035. This plan of action involves measures such as qualified birth care, comprehensive basic antenatal care, management of preterm deliveries and basic neonatal care [6,7].

In the last two decades, several Latin American countries launched or improved their programs on maternal and perinatal health with the purpose to advance the promotion of women's healthcare, sexual and reproductive rights, obstetric care, family planning, abortion, and combat against domestic and sexual violence. These national health programs also have the potential to influence the trends in stillbirths [8,9].

Overall, the stillbirth ratio (per 1,000 livebirths) decreased by 30% in the Latin American and the Caribbean region from 2000 to 2019 (from 11.2 to 7.9, respectively) [10]. However, these estimates may not accurately represent the actual situation in the region. Estimation of stillbirth ratios was based on multiple sources of information such as vital registration systems, medical birth or death registries and health management information systems, nationally representative household surveys with pregnancy histories or reproductive calendars, and population-based studies. Stillbirth ratios and the annual reduction are very heterogeneous on Latin American and Caribbean countries [11]. While Chile (3.1/1,000 livebirths in 2015), Argentina (4.6/1,000 livebirths in 2015) and Uruguay (6.6/1,000 LB in 2015) have shown compelling results in recent years, other countries have shown ratios as high as 24.9/1,000LB (Haiti); 44.0/1,000 livebirths (Bolivia) and 14.8/1,000 livebirths (Brazil) [12]. Understanding the current

burden of stillbirths in the region and the factors associated with its occurrence is ultimately important to reduce heterogeneity and improve related policies in the region.

Valuable information on current challenges is provided when health indicators, such as stillbirth ratios, are monitored in different countries of the region. It can contribute to direct actions to decrease this burden and improve the quality of perinatal health [13,14]. Therefore, this study aimed to determine stillbirth ratios according to maternal and pregnancy characteristics, along with associated factors in some Latin American countries over the last few years.

## Materials and methods

For this study, we used data from the Perinatal Information System (SIP), which is a PAHO/WHO database of the Latin American Center for Perinatology and Women's Reproductive Health (CLAP) [13]. The system provides a standardized electronic record for maternal and neonatal data, with a purposeful tool for both assistance, and research, including data on maternal demographics; family and obstetric history; prenatal visits, delivery, and postpartum details. Information on deliveries was provided from 8 health facilities in five countries participating in the CLAP Maternal and Perinatal Network [14]. For this specific study, the five countries were: Bolivia, Guatemala, Honduras, Nicaragua, and the Dominican Republic.

For analysis, data were extracted from SIP records on maternal and pregnancy characteristics of women who gave birth from August 2018 to June 2021 in those health facilities. Women with a history of abortion, missing information for live births and stillbirths, and incomplete cases were excluded from the analysis. Independent variables considered were age, ethnicity, maternal Education/literacy, marital status, parity, maternal health conditions (including anemia, hypertension, preeclampsia, eclampsia, infections, diabetes, renal disease or cardiac disease). Maternal morbidity according to WHO criteria was also evaluated [15]. These indicators included maternal near-miss (MNM), potentially life-threatening conditions (PLTC), severe maternal outcome (SMO) and maternal death (MD). The conditions related to the definition of maternal morbidity are provided in the supplementary material (S2 Table in S1 File). The occurrence of related conditions was identified by health professionals and they were reported to SIP by local staff assistants. The research team attributed the different degrees of maternal morbidity according to the identification of each condition based on the dataset. To evaluate factors associated with stillbirths according to pregnancy outcomes at each participating center, fetal sex, birth weight, gestational age at birth, and congenital anomalies were the variables used. For confounding factors, placental abruption, mode of delivery, and assistance during childbirth were included. All variables considered in the analysis are listed in detail in S1 Table in S1 File.

The dependent variable stillbirth was defined as a baby born with no signs of life at 28 weeks of gestation or later and birth weight $\geq$1000g [16,17]. The proportion of intrapartum stillbirth, antepartum and ignored time of stillbirth was calculated, considering the time when death was confirmed. The stillbirth rate is defined as the number of fetal deaths per 1,000 total births (live births and fetal deaths), while the stillbirth ratio is the number of fetal deaths per 1,000 live births [16]. For this analysis, we estimated stillbirth ratios for all and each facility [16].

Maternal, pregnancy, and delivery characteristics, along with pregnancy outcomes were reported. All estimates of association were tested for significance using chi-square tests. $P < 0.05$ was regarded as significant. Bivariate analyses were conducted to estimate risk of stillbirth. Prevalence ratios (PR) with their 95% confidence intervals (CI) for each predictor were reported. A backward stepwise logistic regression analysis was performed to assess conditions independently associated with stillbirth. Estimated adjusted prevalence ratios (PRadj) with

their respective 95% CI were reported. The software Stata version 7.0 (StataCorp, College Station, TX, USA) was used for data analysis.

### Ethical aspects

This manuscript is part of the planned products from the "Study on the incidence of severe maternal morbidity and mortality in maternities from the Red-CLAP from Latin America and the Caribbean", approved by the Research Ethics Committee (ERC) from the Pan American Health Organization (PAHO) on Aug 17, 2018 (PAHOERC Ref. No: PAHO-2018-04-0025). The need for a consent form was waived.

### Results

In total, 101,852 childbirths from January 2018 to June 2021 comprised this SIP database. For this analysis, we included 99,712 childbirths. The exclusion of 2,140 childbirths was due to missing information for live birth/stillbirth status (n = 266), abortion (n = 76), multiple pregnancies (n = 1,515) and outliers for gestational age and birth weight (n = 320). During the study period, 762 stillbirths were registered. Of these, 586 (76.9%) were antepartum stillbirths, 150 (19.7%) were intrapartum stillbirths and 26 (3.4%) had an ignored time of death. Fig 1 shows the flowchart of cases considered for analysis. Table 1 demonstrates the prevalence of stillbirths according to participating centers. Individual country ratios ranged from 3.8 per 1,000 livebirths in Guatemala, 4.2 per 1,000 livebirths in Bolivia, 5.8 per 1,000 livebirths in three facilities in Honduras, 7.0 per 1,000 livebirths in two facilities in Nicaragua and in the Dominican Republic that had the highest stillbirth ratio of around 18.2 per 1,000 livebirths.

Table 2 shows stillbirths according to maternal and pregnancy characteristics. The occurrence of stillbirth did not differ by maternal and pregnancy characteristics evaluated, apart from antenatal screening for toxoplasmosis, syphilis, and HIV. Lack of screening for toxoplasmosis, syphilis and HIV during antenatal care was more frequent in women who had stillbirths (3.8% versus 0.9%; p<0.001).

The occurrence of stillbirth according to some maternal health conditions, including maternal morbidity is shown in Table 3. Stillbirth was significantly associated with women with diabetes (PR 4.11; 95%CI [1.73–9.80]), chronic hypertension (PR 2.04; 95%CI [1.45–2.87]), preeclampsia (PR 3.18; 95%CI [2.14–4.72]), eclampsia/HELLP (PR 5.34; 95%CI [1.50–19.06]), renal disease (PR 7.31; 95%CI [3.52–15.19]), and any medical condition (PR 2.39; 95% CI [1.87–3.04]). The risk of stillbirth increased in women who had PLTC (PR 3.41; 95%CI [2.29–5.09]), MNM (PR 11.87; 95%CI [5.73–24.59]), MD (PR 13.81; 95%CI [4.27–44.61]) and SMO (PR 12.25; 95%CI [5.49–27.31]).

Pregnancy outcomes were compared to the occurrence of stillbirths (Table 4). Stillbirths were associated with a higher proportion of placental abruption (PR = 18.59; 95%CI [8.56–40.34]), congenital anomalies (PR = 5.17; 95%CI [1.00–26.86), preterm delivery (gestational age at birth below 32 weeks, PR 47.05; 95%CI [26.72–82.84] or childbirth between 32 and 36 weeks, PR = 7.07; 95%CI [3.59–13.92]) and low birth weight (PR = 14.33; 95%CI [7.23–28.41]). Stillbirth deliveries were less commonly assisted by nurses (PR 0.12; 95%CI [0.06–0.21])

Table 5 addressed independent factors associated with overall stillbirths. Severe maternal outcome (PR$_{adj}$ 6.17; 95%CI [3.27–11.66]), any medical condition (PR$_{adj}$ 1.48; 95%CI [1.24–1.76]), maternal age (years; PR$_{adj}$ 1.04; 95%CI [1.02–1.05]); preeclampsia (PR$_{adj}$ 2.01; 95%CI [1.26–3.19]), ethnicity (not black/pardo (PR$_{adj}$ 0.54; 95%CI [0.34–0.84]) and diabetes (PR$_{adj}$ 2.36; 95%CI [1.25–4.46]) were significantly associated with stillbirth.

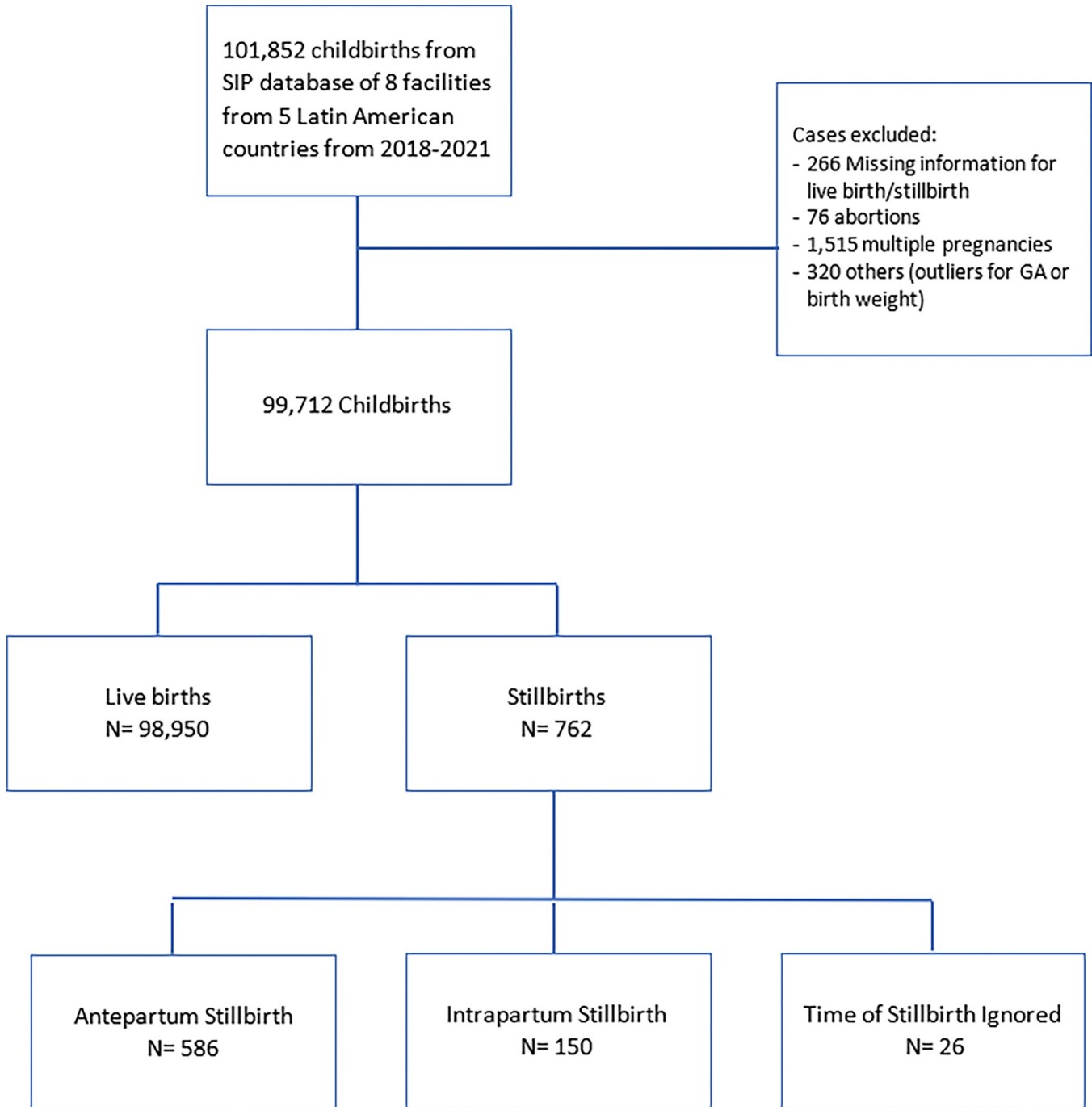

**Fig 1. Flowchart of study participants according to delivery outcomes.** SIP/PAHO 2018–2021.

## Discussion

The present analysis assessed the Perinatal Information System (SIP) database of eight selected obstetric units in five Latin American countries from the CLAP Network to determine stillbirth ratios according to regional, maternal and pregnancy characteristics, in addition to factors associated with stillbirths. Overall, there were 7.7 stillbirths /1,000 livebirths for eight participating countries. The highest ratio occurred in the Dominican Republic (18.2/1,000

**Table 1. Prevalence of stillbirths according to participating centers.** SIP/PAHO 2018–2021.

| Birth outcomes | Total (8 facilities) | Bolivia (1 facility) | Dominican Republic (1 facility) | Guatemala (1 facility) | Honduras (3 facilities) | Nicaragua (2 facilities) |
|---|---|---|---|---|---|---|
| | N (%) | N (%) | N (%) | N (%) | N (%) | N (%) |
| Live births | 98950 (99.2) | 4510 (99.6) | 13810 (98.2) | 11710 (99.6) | 31054 (99.4) | 37866 (99.3) |
| Antepartum Stillbirths | 586 (0.6) | 17 (0.4) | 133 (0.9) | 41 (0.3) | 151 (0.5) | 244 (0.6) |
| Intrapartum Stillbirths | 150 (0.2) | 2 (0.0) | 107 (0.8) | 1 (0.0) | 19 (0.1) | 21 (0.1) |
| Ignored time of stillbirth | 26 (0.0) | 0 (0.0) | 12 (0.1) | 3 (0.0) | 10 (0.0) | 1 (0.0) |
| Stillbirth ratio (/1,000 livebirths) | 7.7 | 4.2 | 18.2 | 3.8 | 5.8 | 7.0 |

livebirths) and the lowest in Guatemala (3.8/1000 livebirths). Stillbirth was significantly associated with women with diabetes, preeclampsia, maternal age, ethnicity, any medical condition, and severe maternal outcome.

Latin America is a region with distinct characteristics related to economic, social and healthcare inequalities There are significant differences in health outcomes based on the population's wealth, geographic location, education, age, and ethnicity [18]. It is recognized as the most unequal region in the world, and 24% of the population (around 142 million people) used to live in poverty in 2013 [18]. Children, adolescents, and women remain the most vulnerable population in the region [19]. Major risk factors independently associated with stillbirths in our results were related to maternal sociodemographic and pregnancy characteristics e.g. age, ethnicity, and maternal morbidity. These findings are consistent with previous studies held in higher-income countries. A substantial relationship exists between stillbirths and access (to) and quality of antenatal care, which has a key role in the prevention of stillbirths [20–23]. Decreasing access barriers in health care is crucial to reduce preventable deaths, especially through adequate surveillance and management of maternal complications. This is remarkably important not only for women with chronic comorbidities but also for all those who have a pregnancy-related maternal complication.

The estimated stillbirth ratio in 2015 was 8.1 per 1000 live births in Latin America, with a wide variation among countries [20]. Progress was noted, when data from 2000 to 2019 [11] were compared. However, structural and social inequalities remain a major issue in Latin American countries. Special attention is needed for adequate management, improvement in obstetric surveillance, to promote a decline in fetal death ratio in the region.

Hypertensive disorders of pregnancy are frequent and represent an important complication of pregnancy. It is a major cause of maternal and perinatal morbidity and mortality, especially in preeclampsia/eclampsia patients, in low-income countries [24]. In a large WHO trial, involving seven developing countries, hypertensive disorders of pregnancy were responsible for 28% of stillbirths, following prematurity as the most common cause of stillbirth [25]. Stillbirths in pregnancies complicated by hypertensive disorders are commonly associated with abruptio placentae, uteroplacental insufficiency and placental infarctions [26]. Our results confirmed the association between placental abruption and stillbirth, recognizing the dramatic outcome followed by such complication. Strict monitoring of pregnant women with careful prenatal care is crucial to reduce maternal and perinatal mortality, e.g., assessment of fetal vitality by Doppler velocimetry is undoubtedly very useful in pregnancies that are at a higher risk for morbidity and mortality [27].

Technically, diabetes mellitus affects 2% to 5% of all pregnancies. Most stillbirths occur in the third trimester to patients with poor glycemic control. Diabetes can lead to complications

**Table 2. Stillbirth according to maternal and pregnancy characteristics.** SIP/PAHO 2018–2021.

| Characteristics | Stillbirth N (%) | Live Birth N (%) | p-value |
|---|---|---|---|
| **Maternal age (years)[a]** | | | <0.001 |
| ≤19 | 159 (21.6) | 24937 (25.5) | |
| 20–34 | 459 (62.8) | 65082 (66.7) | |
| ≥35 | 113 (15.5) | 7627 (7.8) | |
| **Ethnicity/Skin color[b]** | | | 0.079 |
| Black/Pardo | 683 (95.0) | 89394 (92.2) | |
| White | 4 (0.6) | 597 (0.6) | |
| Indigenous | 27 (3.8) | 6173 (6.4) | |
| Others | 5 (0.7) | 840 (0.9) | |
| **Literacy[c]** | | | 0.191 |
| None or primary | 289 (40.6) | 35376 (36.6) | |
| Secondary or University | 423 (59.4) | 61158 (63.4) | |
| **Marital status[d]** | | | 0.236 |
| Married/stable part | 647 (92.8) | 89484 (94.2) | |
| Single/Other | 50 (7.2) | 5527 (5.8) | |
| **Parity[e]** | | | 0.064 |
| Nulliparous | 278 (38.1) | 40781 (41.8) | |
| Multipara | 452 (61.9) | 56791 (58.2) | |
| **Previous stillbirths [f]** | | | 0.136 |
| Yes | 10 (2.4) | 764 (1.5) | |
| No | 411 (97.6) | 51827 (98.5) | |
| **ANC screening for Toxoplasmosis, Syphilis, and HIV [g]** | | | <0.001 |
| Done | 100 (13.1) | 11060 (11.2) | |
| Partially done | 633 (83.1) | 87013 (87.9) | |
| Not done | 29 (3.8) | 877 (0.9) | |
| **Toxoplasmosis (ANC)[h]** | | | 0.129 |
| Yes | 23 (21.3) | 2952 (26.0) | |
| No | 85 (78.7) | 8407 (74.0) | |
| **Syphilis (ANC)[i]** | | | 0.056 |
| Yes | 11 (1.7) | 588 (0.6) | |
| No | 640 (98.3) | 92015 (99.4) | |
| **HIV (ANC)[j]** | | | 0.831 |
| Yes | 3 (0.4) | 452 (0.5) | |
| No | 729 (99.6) | 97595 (99.5) | |

*Missing information for a: 1,335; b:1,989; c: 2,466; d: 4,004; e: 1,410; f: 4,230; g: 0,00; h: 88,245; i: 6,458; j: 933. ANC: antenatal care.

such as macrosomia, polyhydramnios, intrauterine fetal growth restriction, and preeclampsia [28]. Pre-gestational diabetes with attendant vascular complications is a significant risk factor for stillbirth, which highlights the importance of adequate metabolic control before and during pregnancy [26]. Relevant guidelines such as the PAHO Managing Diabetes in Primary Care in the Caribbean have been implemented to improve the quality of care in diabetes mellitus. The protocol emphasizes the importance of intensive antenatal surveillance, maintenance of euglycemia, fetal surveillance, and provision of timely delivery. All these measures have been shown to decrease the risks of intrauterine fetal demise [29,30]. Therefore, it is extremely important to implement effective preventive strategies, such as encouraging women to consume a healthy

Table 3.  Stillbirths according to some maternal health conditions.  SIP/PAHO 2018–2021.

| Maternal health conditions | Live birth (n = 98950) | Stillbirth (n = 762) | |
|---|---|---|---|
| | N (%) | N (%) | PR (95%CI) |
| **Medical conditions** | | | |
| Anemia [a] | 7664 (7.9) | 86 (11.9) | **1.57 (1.20–2.06)** |
| Diabetes [b] | 549 (0.6) | 17 (2.3) | **4.11 (1.73–9.80)** |
| Hypertension [c] | 1791 (1.8) | 27 (3.7) | **2.04 (1.45–2.87)** |
| Preeclampsia [d] | 5280 (5.4) | 112 (15.6) | **3.18 (2.14–4.72)** |
| Eclampsia/HELLP [e] | 845 (0.9) | 33 (4.6) | **5.34 (1.50–19.06)** |
| Cardiac disease [f] | 327 (0.3) | 3 (0.4) | 1.22 (0.37–4.06) |
| Renal disease [g] | 105 (0.1) | 6 (0.8) | **7.31 (3.52–15.19)** |
| Other severe medical conditions [h] | 578 (0.6) | 2 (0.3) | 0.47 (0.22–1.01) |
| Any medical condition [i] | 14629 (15.1) | 212 (30.0) | **2.39 (1.87–3.04)** |
| **Continuum of Maternal morbidity [j]** | | | |
| No morbidity | 82948 (93.5) | 490 (76.8) | Ref. |
| PLTC | 5231 (5.9) | 107 (7.8) | **3.41 (2.29–5.09)** |
| MNM | 427 (0.5) | 32 (5.0) | **11.87 (5.73–24.59)** |
| MD | 102 (0.1) | 9 (1.4) | **13.81 (4.27–44.61)** |
| SMO | 529 (0.6) | 41 (6.4) | **12.25 (5.49–27.31)** |

*PTLC: Potentially life-threatening condition; MNM: Maternal near-miss; SMO: severe maternal outcome (MNM or maternal death); MD: maternal death.
Missing information for a: 1571; b: 938; c: 929; d: 1401; e: 1381; f:770; g: 766; h: 1,276; i: 2,204; j: 10,366.

diet and providing management protocols for diabetic pregnant women [31,32]. Maintenance of blood sugar control should be ensured to optimize perinatal outcomes.

Infection is an important cause of stillbirths worldwide. Around 50% of stillbirths are probably caused by infection during pregnancy in low- and middle-income countries [33,34]. Gestational syphilis is considered a major contributing factor for stillbirth in the Americas, increasing six times the risk of stillbirth [35]. Therefore, antenatal tests are important to identify and provide proper care to both mother and fetus. In our study, the risk of stillbirth was higher in women who did not test for toxoplasmosis, syphilis and HIV. We assume that this association is primarily due to inadequate antenatal and obstetric care services. Our findings reinforce the importance of providing proper antepartum and intrapartum care and confirm the importance of infection prevention and management for this particular higher-risk population.

Training and support for healthcare workers, strengthening of health systems; and counseling for pregnant women, are examples of investments to prevent stillbirth [7,9]. Improving stillbirth reporting in routine information systems to audit and classify perinatal mortality in the region would help strengthen the organizational network and also bring stillbirth out of the shadows, achieving the target of ending preventable stillbirths by 2030, maternal and neonatal deaths as presented in the Lancet Ending Preventable Stillbirth Series [36]. Furthermore, adequate use of already existing interventions and surveillance of related indicators have a great potential to prevent stillbirths in Latin American and Caribbean countries [37]. The use of a Perinatal Information System such as SIP can be a resourceful tool to promote reliable surveillance of the provision of health care interventions.

Experiencing the birth of a stillborn child is life-changing. Stillbirth is associated with various adverse psychosocial outcomes in parents [38]. Stress, anxiety, and depressive symptoms are common immediate reactions to stillbirth. Negative psychological effects of the loss may

**Table 4. Delivery characteristics according to pregnancy outcomes.** SIP/PAHO 2018–2020.

| Pregnancy outcomes | Live birth (n = 98950) | Stillbirth (n = 762) | |
|---|---|---|---|
| | N (%) | N (%) | PR(95%CI) |
| **Placental abruption [a]** | | | |
| Yes | 201 (0.2) | 31 (4.4) | 18.59 (8.56–40.34) |
| No | 92663 (99.8) | 671 (95.6) | Ref. |
| **Congenital anomalies [b]** | | | |
| Yes | 955 (1.0) | 32 (5.1) | **5.17 (1.00–26.86)** |
| No | 94481 (99.0) | 596 (94.9) | Ref. |
| **Mode of delivery [c]** | | | |
| C-section | 36349 (37.0) | 204 (27.3) | Ref. |
| Spontaneous vaginal | 61770 (62.9) | 541 (72.5) | 1.56 (0.98–2.46) |
| Operative vaginal | 155 (0.2) | 1 (0.1) | 1.15 (0.07–18.90) |
| **Assistance during childbirth [d]** | | | |
| MD /obstetrician | 80844 (85.6) | 626 (89.3) | Ref. |
| Nurse | 3376 (3.6) | 3 (0.4) | 0.12 (0.06–0.21) |
| Only other health professionals | 10218 (10.8) | 72 (10.3) | 0.91 (0.31–2.67) |
| **Sex of the child births [e]** | | | |
| Male | 50912 (51.7) | 386 (52.6) | Ref. |
| Female | 47514 (48.3) | 348 (47.4) | 1.03 (0.95–1.13) |
| **Gestational age at birth [f]** | | | |
| <32 weeks | 1125 (1.2) | 218 (31.8) | **47.05 (26.72–82.84)** |
| 32–36 | 6400 (6.6) | 160 (23.3) | **7.07 (3.59–13.92)** |
| ≥ 37 weeks | 88959 (92.2) | 308 (44.9) | Ref. |
| **Birth weight [g]** | | | |
| <2500g | 8146 (8.3) | 426 (57.1) | **14.33 (7.23–28.41)** |
| 2500-3999g | 87955 (89.2) | 306 (41.0) | Ref. |
| ≥4000g | 2458 (2.5) | 14 (1.9) | 1.63 (0.57–4.69) |

*Missing information for a: 6,146; b: 3,648; c: 692; d: 4,573; e: 552; f: 2,542; g: 407.

continue into subsequent pregnancies, despite the birth of a healthy child. Therefore, preventive actions, such as professional and social support, can facilitate a healthy grieving process and reduce the risk of developing lasting pathological symptoms [38,39]. Most importantly, the investigation and identification of the causes of fetal death would allow management systematization, an action that could reduce the prevalence of fetal death, decreasing its risk in subsequent pregnancies.

The current study has strengths and limitations. In low-income countries, a better understanding of pregnancy outcomes is hindered by the scarcity of reliable vital statistics and lack of consistency in available data over time or between geographic areas. Although our study was based on a large dataset of the SIP database, the numbers do not represent the reality of each country or region. A facility-based electronic health information system containing perinatal data is purposeful for local surveillance, but spreading its implementation is crucial for promoting consistent and representative regional data. Part of the variability in stillbirth ratios could be attributed to a potential variation in definition, registration and verification of stillbirths across different facilities. For international comparability, we used the World Health Organization (WHO) definition of stillbirth as fetal death at or after 28 weeks gestation or ≥1000g birth weight prior to complete fetal expulsion or extraction from the mother [16,17].

**Table 5. Variables independently associated with stillbirth.** SIP/PAHO 2018–2021[n = 88,037].

| Variables | PR$_{adj}$ | 95% CI | p-value |
|---|---|---|---|
| SMO (MNM+MD) | 6.17 | (3.27–11.66) | <0.001 |
| Any medical condition | 1.48 | (1.24–1.76) | 0.001 |
| Maternal age (years) | 1.04 | (1.02–1.05) | 0.002 |
| Preeclampsia | 2.01 | (1.26–3.19) | 0.009 |
| Ethnicity (not Black/Pardo) | 0.54 | (0.34–0.84) | 0.013 |
| Diabetes | 2.36 | (1.25–4.46) | 0.015 |

PR$_{adj}$ = Prevalence Ratio adjusted for significant predictors and cluster design (eight facilities). CI confidence interval for PR. Predictors entering the first model: Maternal age (years); Ethnicity (Black/Pardo:0/ Not Black/Pardo:1); Literacy (No or primary:1/ Secondary or university:0); Marital status (Married+ stable part:0/ Single+other:1); Parity (0/ >0:1); Syphilis (Yes:1/ No, not done:0); Onset of labor: Spontaneous/ Induced+Elective C-section); Anemia (Yes:1/ No:0); Diabetes (Yes:1/ No:0); Hypertension (Yes:1/ No:0); Preeclampsia (Yes:1/ No:0); Eclampsia/HELLP (Yes:1/ No:0); Cardiac disease (Yes:1/ No:0); Renal disease (Yes:1/ No:0); Other severe medical condition (Yes:1/ No:0); SMO (MNM+MD:1/ No morbidity+PLTC:0); Mode of delivery (Spontaneous vaginal:0/ C-section+operative vaginal:1); Assistance during childbirth (MD+obstetrician:0/ Nurse+only other health professionals:1); Congenital anomalies ('Major':1/ No+'minor':0); Neonatal reanimation required (Yes:1/ No:0).

Although network health facilities have used WHO´s definitions of a stillbirth (baby born with no signs of life at 28 weeks of gestation or more, and birth weight ≥1000g [16]), routinely collected data may encompass minor errors. Limitations of facility-based and routinely collected data include a lack of sensitivity of some indicators based on outpatient data (e.g., delivery outside the obstetric unit) and variation in data quality that is dependent on local standards and audit frequency. Another limitation is that we did not address causes attributed to the stillbirth cases. Although the WHO has proposed a standardized approach for investigating and reporting fetal deaths, the SIP does not contain the cause attributed to the fetal death in the data registry [17]. Including the cause of fetal deaths in the SIP could contribute to audit the implementation of such standardized audit and review of fetal deaths. Implementing a standard classification system and post-mortem investigation would be of great benefit for assessing causes and informing root-cause of deaths, potentially decreasing the proportion of unspecified and unspecific causes and informing valuable information for preventive strategies [15,40–42]. There are different classifications systems for fetal death causes such as the Aberdeen, Wigglesworth, Recode, Tulip, CODAC, ICD-PM and a structured system developed by the Stillbirth Collaborative Research Network [19,20,43,44]. Applying such classification systems promotes a standardized determination of causes of death based on clear criteria and the identification associated factors [40].

## Conclusions

Pregnancy complications and maternal morbidity were significantly associated with stillbirths. Specialized antenatal and intrapartum care remains a priority, particularly for women at a higher risk for stillbirth. Furthermore, optimal management includes counselling about the importance of screening tests in pregnancy and easily accessible medical care for these associated conditions.

## Supporting information

**S1 Checklist. STROBE statement—checklist of items that should be included in reports of observational studies.**
(DOCX)

**S1 File. S1 Table.** Independent variables considered in the analysis. S2 Table. The WHO criteria for potentially life-threatening conditions and maternal near miss.
(DOCX)

## Acknowledgments

We thank all professionals who directly or indirectly participated in the data collection for this analysis and the PAHO-CLAP support for that. We also are grateful for the important input of Pablo Duran from PAHO-CLAP to the review of the manuscript.

## Author Contributions

**Conceptualization:** Bremen de Mucio, Claudio Sosa, Mercedes Colomar, Rigoberto Castro, Sherly Metelus, Renato T. Souza, Maria L. Costa, Adriana G. Luz, Maria H. Sousa, José G. Cecatti, Suzanne J. Serruya.

**Data curation:** Bremen de Mucio, Claudio Sosa, Mercedes Colomar, Luis Mainero, Carmen M. Cruz, Luz M. Chévez, Rita Lopez, Gema Carrillo, Ulises Rizo, Erika E. Saint Hillaire, William E. Arriaga, Rosa M. Guadalupe Flores, Carlos Ochoa, Freddy Gonzalez, Rigoberto Castro, Allan Stefan, Amanda Moreno, Suzanne J. Serruya.

**Formal analysis:** Sherly Metelus, Renato T. Souza, Maria H. Sousa, José G. Cecatti, Suzanne J. Serruya.

**Investigation:** Bremen de Mucio, Claudio Sosa, Mercedes Colomar, Luis Mainero, Carmen M. Cruz, Luz M. Chévez, Rita Lopez, Gema Carrillo, Ulises Rizo, Erika E. Saint Hillaire, William E. Arriaga, Rosa M. Guadalupe Flores, Carlos Ochoa, Freddy Gonzalez, Rigoberto Castro, Allan Stefan, Amanda Moreno, Sherly Metelus, Renato T. Souza, Maria L. Costa, Adriana G. Luz, José G. Cecatti, Suzanne J. Serruya.

**Methodology:** Bremen de Mucio, Claudio Sosa, Mercedes Colomar, Luis Mainero, Carmen M. Cruz, Luz M. Chévez, Rita Lopez, Gema Carrillo, Ulises Rizo, Erika E. Saint Hillaire, William E. Arriaga, Rosa M. Guadalupe Flores, Carlos Ochoa, Freddy Gonzalez, Rigoberto Castro, Allan Stefan, Amanda Moreno, Sherly Metelus, Renato T. Souza, Maria L. Costa, Adriana G. Luz, José G. Cecatti, Suzanne J. Serruya.

**Project administration:** Luis Mainero, Suzanne J. Serruya.

**Resources:** Claudio Sosa, Suzanne J. Serruya.

**Software:** Luis Mainero, Suzanne J. Serruya.

**Supervision:** Luis Mainero, José G. Cecatti.

**Visualization:** Bremen de Mucio.

**Writing – original draft:** Bremen de Mucio, Sherly Metelus, Renato T. Souza.

**Writing – review & editing:** Bremen de Mucio, Claudio Sosa, Mercedes Colomar, Luis Mainero, Carmen M. Cruz, Luz M. Chévez, Rita Lopez, Gema Carrillo, Ulises Rizo, Erika E. Saint Hillaire, William E. Arriaga, Rosa M. Guadalupe Flores, Carlos Ochoa, Freddy Gonzalez, Rigoberto Castro, Allan Stefan, Amanda Moreno, Sherly Metelus, Renato T. Souza, Maria L. Costa, Adriana G. Luz, Maria H. Sousa, José G. Cecatti, Suzanne J. Serruya.

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
