## [Decision Letter · Decision Letter 0]

10 Jul 2023

PONE-D-23-05294The burden of stillbirths in low resource settings in Latin America: evidence from a network using an electronic surveillance systemPLOS ONE

Dear Dr. Jose G Cecatti,

Thank you for submitting your manuscript to PLOS ONE. After careful consideration, we feel that it has merit but does not fully meet PLOS ONE’s publication criteria as it currently stands. Therefore, we invite you to submit a revised version of the manuscript that addresses the points raised during the review process.

Dear Dr. Jose G Cecatti,

Thank you for submitting your manuscript to PLOS ONE. After careful consideration, we feel that it has merit but does not fully meet PLOS ONE’s publication criteria as it currently stands. Therefore, we invite you to submit a revised version of the manuscript that addresses the points raised during the review process.

A native English speaker should proofread the paper.  Regarding the definition of a stillbirth and another part of your manuscript, reviewers highlighted concerns.

We look forward to receiving your revised manuscript.

Journal Requirements: 

Reviewers' comments:

Reviewer's Responses to Questions

**Comments to the Author**

1. Is the manuscript technically sound, and do the data support the conclusions?

Reviewer #1: Partly

Reviewer #2: Yes

2. Has the statistical analysis been performed appropriately and rigorously? 

Reviewer #1: Yes

Reviewer #2: Yes

3. Have the authors made all data underlying the findings in their manuscript fully available?

Reviewer #1: Yes

Reviewer #2: No

4. Is the manuscript presented in an intelligible fashion and written in standard English?

Reviewer #1: Yes

Reviewer #2: Yes

5. Review Comments to the Author

Reviewer #1: Thank you for this article, which provides us with very valuable information on stillbirth rates, risk factors and regional differences of five Latin American countries. The authors analyzed a large dataset of almost 100000 births.

General comments

- A native English speaker should proofread the article

- Text must be more concise and objectives stated more clearly

- Focus on the flow of the article and apply more structure in the text

- With the intention to compare data, I would advise to use ICD-PM and the definition of the WHO

Comments

1. Abstract

- Are we reporting stillbirth rates (line 33) or ratio line 46) ? The WHO uses as denominator live births and stillbirths ( ref https://www.who.int/docs/default-source/mca-documents/maternal-nb/making-every-baby-count.pdf.). Please also read https://www.ncbi.nlm.nih.gov/pmc/articles/PMC6631739/ on how to define stillbirths and what to use as denominator, don’t use rate and ratio’s as they are not the same.

- In the objective in the abstract line 33 -34 the authors state that they determine the stillbirth rate and the regional, maternal end pregnancy characteristics and associated factors. Do they mean risk factors associated with stillbirths?

- Be more specific in what exactly your objectives are ( -determine stillbirth rate, analyze the association with maternal, perinatal and delivery characteristics and geographical differences?)

- In the abstract we don’t see any conclusion about the regions. If you state it in the objective, you have to say something about it in the result/conclusion

- In line 36 please add SIP as abbreviation for the first time

2. Introduction

- line 66-67 please correctly use what definition the reference uses – 22 weeks and not 20 weeks is the threshold – early and late stillbirth definition.

- Start the introduction with relevant information about the worldwide burden is of stillbirth (first paragraph) and than continue to Latin America. Now the first 4 lines is only about the definition and this drags the attention away from the article

- Line 72 uses now FDR – it is better to not use several terms like rate and ratios stillbirth and fetal death without giving explanation…. Please read the article https://www.ncbi.nlm.nih.gov/pmc/articles/PMC6631739/ on how to define stillbirths and what to use as denominator, don’t use rate and ratio’s as they are the same.

- Line 78 “3 million maternal, fetal and neonatal deaths….” Is unclear what is 3 million….

- Be aware to be consistent in the definition of stillbirth rate (WHO – denominator is the total births)

- Line 97-101 is too long for one sentence and by referring to the article you can summarize what you want to say – stillbirth rates and the annual reduction are very heterogenous on … and then summarize .. because all the figures in the introduction makes it too long.

- Relate these differences in the stillbirth rate in the Latin American-Caribbean region with the objective of your study so that we can better understand the relevance of this study.

- Line 104– is it contributing or could contribute? Are the countries doing this now? Do you have a reference than where individual countries do that?

3. Material and methods

- Please use the strobe checklist and add more structure to the methods section and provide all information accordingly such as describe the type of study etc. https://www.strobe-statement.org/checklists/

- Line 121-142 can be made better understandable with a flowchart or table focusing on 1. Maternal characteristics ( age, ethnicity….) 2. pregnancy characteristics (…..), 3. delivery characteristics (…..)

- Line 146 – stillbirth ratio – and total live birth is used as denominator…. This should be more specific and comparable – so please use WHO criteria so that the data can be compared!

- I would suggest, although you don’t have the causes of death to use the ICD-PM to be able to compare data ( https://pubmed.ncbi.nlm.nih.gov/35972943/ , https://pubmed.ncbi.nlm.nih.gov/32777997/ ).

- Maternal morbidity (MNM, PTLC, SMO, MD) : who attributed maternal morbidity to a case? Who defined a case as a MNM, PTLC etc? and how accurate is the database on this, who classified the case as …..What is the reason the authors choose for this (MNM, PTLC, SMO, MD)

- Is it possible to group the maternal condition according to ICD-PM groups M1-M5

4. Results

- Line 172, do not mention definition again

5. Discussion

- Mention your main results, after line 209 also report the other findings shortly

- Discuss than those findings the authors started with geographical differences, try to explain why some countries might have higher stillbirth rates….

- Than discuss the relevant maternal (hypertension, diabetes, infection) pregnancy and delivery outcomes (as is

- Line 255-260 – we do not have any results on BMI, in the discussion discuss your own findings….

- One limitation is that there are no causes attributed to the stillbirth cases

- Line 313 -314 the conclusion that the SIP is a tool of utmost importance is not in the scope of this study

Reviewer #2: The authors discuss a very important subject in maternal health and will benefit the readers in many settings.

A few comments for the authors to clarify

Abstract

Line 44 Authors use the acronym SIP without defining it first.

Main manuscript

Methods

Line 154 The authors state 95% Cis instead of CIs

Results

Do the 8 participating facilities receive information from other sites in their countries? If so, how does information flow? Still birth rates are quoted per country. How representative are these facilities of what is happening nationally?

Is blood sugar a routine test in antenatal care in these countries?

6. PLOS authors have the option to publish the peer review history of their article (what does this mean?). If published, this will include your full peer review and any attached files.

Reviewer #1: No

Reviewer #2: No

---

## [Author Response · Author response to Decision Letter 0]

25 Jul 2023

the Reply letter is uploades as cover letter

---

## [Decision Letter · Decision Letter 1]

5 Dec 2023

The burden of stillbirths in low resource settings in Latin America: evidence from a network using an electronic surveillance system

PONE-D-23-05294R1

Dear Dr. Jose G Cecatti, We’re pleased to inform you that your manuscript has been judged scientifically suitable for publication and will be formally accepted for publication once it meets all outstanding technical requirements.

Kind regards,

Hassen Mosa, Msc

Academic Editor

PLOS ONE

Additional Editor Comments (optional):

Reviewers' comments:

Reviewer's Responses to Questions

**Comments to the Author**

1. If the authors have adequately addressed your comments raised in a previous round of review and you feel that this manuscript is now acceptable for publication, you may indicate that here to bypass the “Comments to the Author” section, enter your conflict of interest statement in the “Confidential to Editor” section, and submit your "Accept" recommendation.

Reviewer #2: All comments have been addressed

Reviewer #3: (No Response)

2. Is the manuscript technically sound, and do the data support the conclusions?

Reviewer #2: Yes

Reviewer #3: (No Response)

3. Has the statistical analysis been performed appropriately and rigorously? 

Reviewer #2: Yes

Reviewer #3: (No Response)

4. Have the authors made all data underlying the findings in their manuscript fully available?

Reviewer #2: Yes

Reviewer #3: (No Response)

5. Is the manuscript presented in an intelligible fashion and written in standard English?

Reviewer #2: Yes

Reviewer #3: (No Response)

6. Review Comments to the Author

Reviewer #2: The authors have adequately addressed the comments raised. The authors refer readers to a third party for data used in the study through an email address. Language has improved although very few typos (double full stops, random capitalization) are present.

Reviewer #3: I would like to thank authors for their revision. Author has made a significant improvement in their revision by addressing the comments/suggestions from prior reviewers, however, I would suggest incorporating the following specific comments that could improve the readability and quality of the manuscript.

1) The objective of the study in the abstract needs to be simplified like “to determine stillbirth ratio and its association with maternal, perinatal, and delivery characteristics, as well as geographic differences in Latin American countries (LAC)”.

2) In the result section of the abstract, you should incorporate country-specific stillbirth ratio as well, to be aligned with your objective focused on geographic differences. Also, you need to revise your conclusion accordingly. As you did not perform your analysis based on the maternity hospitals, therefore, you cannot conclude that the stillbirth ratios varied across the maternity hospitals, rather say- varied across countries.

3) In main text, you used an abbreviation of live birth as LB, which is not commonly used. I would suggest keeping its full form like live birth, which would be easy to understand for readers.

4) Could you please provide a clear script of STATA code (do file) as an appendix? I am surprised to see that author used a quite older version of STATA (version 7.0) for their analysis. I am wondered if author had any typo (e.g., STATA version 17.0???) as the whole manuscript has several typo- or grammatical errors like 95% Cis or required space/comma between words in some places. Please thoroughly check all types of inconsistencies in sentences into the whole manuscript.

5) In table 4, you named a variable “sex” under the pregnancy outcomes seemed to be “Sex of the child births”. Could you please confirm it and rephrase as suggested.

7. PLOS authors have the option to publish the peer review history of their article (what does this mean?). If published, this will include your full peer review and any attached files.

Reviewer #2: No

Reviewer #3: **Yes: **Dr Md. Obaidur Rahman

---

## [Editor Report · Acceptance letter]

13 Dec 2023

PONE-D-23-05294R1 

PLOS ONE

Dear Dr. Cecatti, 

I'm pleased to inform you that your manuscript has been deemed suitable for publication in PLOS ONE. Congratulations! Your manuscript is now being handed over to our production team.

Kind regards, 

on behalf of

Mr Hassen Mosa 

Academic Editor

PLOS ONE